# Effects of Artificial Light at Night on Fitness-Related Traits of Sea Urchin (*Heliocidaris crassispina*)

**DOI:** 10.3390/ani13193035

**Published:** 2023-09-27

**Authors:** Xiuwen Xu, Zexianghua Wang, Xiuqi Jin, Keying Ding, Jingwen Yang, Tianming Wang

**Affiliations:** National Engineering Laboratory of Marine Germplasm Resources Exploration and Utilization, Marine Science and Technology College, Zhejiang Ocean University, Zhoushan 316022, China; xiuwenxu1207@163.com (X.X.); wangzexianghua@zjou.edu.cn (Z.W.); 2021060@zjou.edu.cn (X.J.); dingkeying1228@163.com (K.D.); silence84309@163.com (J.Y.)

**Keywords:** ALAN, sea urchin, fitness, behavior, GSI, growth

## Abstract

**Simple Summary:**

The sea urchin (*Heliocidaris crassispina*) is an ecologically important invertebrate in structuring marine benthic communities. Most dwell within intertidal regions, rendering them highly susceptible to elevated artificial light at night. However, their potential acclimation to artificial light exposure remains largely unexplored. This study investigates the changes of prolonged artificial light pollution on the fitness-related traits of sea urchins. Following a six-week exposure to artificial light at night, survival remained unaffected, while behavioral responses exhibited slower reactions. Concurrently, growth inhibition was observed in sea urchins, which might be attributed to reduced mouthparts weight and decreased food consumption. This study also revealed that sea urchin gonads are more susceptible to artificial light than guts. *Pax6* gene expression may serve as a sensitive indicator to assess the impact on the photosensitive system of sea urchins. These results increase our understanding of the effects of artificial light at night on sea urchins and provide valuable information about coastal animals’ safety.

**Abstract:**

Limited data are available regarding the effects of elevated coastal artificial light at night (ALAN) on intertidal echinoderms. In this study, we investigated the behavioral, morphological, and physiological responses of the sea urchin (*Heliocidaris crassispina*) after continuous exposure to ALAN at light intensities of 0.1, 300, and 600 Lux for 6 weeks. Our findings revealed that ALAN at 300 Lux substantially reduced food consumption, Lantern weight, and gonadosomatic index (GSI). On the other hand, ALAN at 600 Lux notably prolonged the righting and covering response times and elevated the 5-HIAA/5-HT ratio, while concurrently decreasing food consumption, body weight, Lantern weight, GSI, and *Pax6* gene expression. These results indicated that continuous exposure to ALAN could cause an adverse effect on fitness-related traits, including behavioral responses, growth, reproductive performance, and photoreception of sea urchins. The present study provides new insights on the impact of light pollution on echinoderms.

## 1. Introduction

Artificial light at night (ALAN) changes the natural state of a dark environment from dark to bright by increasing the distribution and intensity of light in space [1,2,3]. As coastal population densities surge, the proliferation of ALAN is expanding at an annual growth rate of approximately 6% [2,4]. While ALAN offers substantial convenience for human activities, it concurrently appears to present a potential threat to coastal ecosystems [2,5,6,7,8,9]. For example, in shallow water coral reef ecosystems, recent studies reported the gametogenesis cycle of *Acropora millepora* was delayed or masked by exposure to ALAN [10]. In sandy beach ecosystems, sea turtle orientation was negatively affected by ALAN, impairing the ability of hatchlings to respond to natural orientation cues [11,12]. In seabird communities, increasing ALAN levels in and around nesting colonies could impact sea mew breeding behavior and, consequently, chick provisioning [13]. However, scant attention has been directed towards investigating the implications of ALAN in intertidal ecosystems, particularly concerning invertebrates such as slow-moving echinoderms. Therefore, a comprehensive grasp of how intertidal invertebrates’ fitness reacts to ALAN exposure is imperative as it provides essential insights into potential ecological risks in coastal environments.

Sea urchin *Heliocidaris crassispina* (previously *Anthocidaris crassispina*) is an echinoderm species found across a wide range, from the intertidal zone to shallow waters in the Western Pacific region. Its habitat spans from the Japan Sea (at 38 °N) to Hainan Island (at 19 °N) [14,15,16,17]. This species holds a crucial role in intertidal ecosystems, acting both as an herbivorous grazer on macroalgal species such as *Ulva Lactuca* and as prey for crabs and starfish [18,19]. Echinoderm behaviors (such as righting, covering, and feeding) are important for their fitness but also affect intertidal ecosystems through effects on their predators, prey, and competitors [20]. Righting is the behavior of inverted sea urchins placed on their aboral surface to resume the posture with the aboral side up [21,22]. Covering behavior refers to sea urchins using their tubular feet and spines to move objects such as algae fragments, stones, and shells to the dorsal surface [23,24,25]. These two behaviors are used to help sea urchins defend against abiotic disturbances and predation [24]. Health-related attributes, such as food consumption, body weight, and gut-weight, alongside reproductive traits like gonad weight, constitute foundational elements influencing the fitness of sea urchins [20,21]. Serotonin serves as one of the monoamine neurotransmitters integral to regulating animal behavior [26]. It can be used to reflect the physiological state of sea urchins when their behavioral expression changes. *Pax6* is a pivotal opsin transcription factor gene within the animal photosensitive system. Deviations in *Pax6* gene expression could indicate a disruption of the light defense system in sea urchins [27].

The primary objective of this study is to examine the effects of prolonged artificial lighting on the righting and covering behaviors, health-related and reproductive traits, coelomic monoaminergic activity, and *Pax6* gene expression in *H. crassispina*. This investigation aims to offer novel insights into the repercussions of ALAN on intertidal creatures, alongside potential underlying mechanisms.

## 2. Materials and Methods

Before commencing the study, we meticulously reviewed animal experiments and welfare regulations, ensuring strict adherence. All procedures undertaken in this study received approval from the Institutional Animal Care and Use Committee of Zhejiang Ocean University.

### 2.1. Sea Urchin Materials

Sea urchins *H. crassispina* were collected from the waters near Zhoushan Islands (122°41’27.040” E, 30°12’41.625” N) by scuba diving and delivered to a 5000 L PVC tank in the Culture Enhancement Laboratory of Zhejiang Ocean University for acclimation. The PVC tank was supplied with filtered seawater at constant aeration and a flow rate of 10 L/min. The seawater was changed every 5 days and its temperature was set at 24.5–26.5 °C. Daytime lighting was provided by low-pressure mercury discharge fluorescent lamps (626–1023 Lux, 12 h light-12 h dark). Sea urchins were fed ad libitum with chopped *U. Lactuca* and their droppings were cleaned every day [21,28].

### 2.2. Experimental Design

After 3 weeks of acclimation, seventy-two sea urchins were selected from the PVC tank and weighed for this experiment (body weight ± SD =54.402 ± 6.240 g, *n* = 72). The weighed sea urchins were then randomly divided evenly into nine 150 L glass tanks (long × wide × deep = 60 cm × 50 cm × 50 cm) placed in the laboratory (*n* = 8 for each tank; three tanks for each group; *n* = 24 for each group). The glass tanks’ condition during the experiment was similar to the PVC tank except for artificial light illumination at night. Nocturnal illumination stimulation was provided by metal halide streetlights (HC-ZY521, OSRAM, Munich, Germany) mounted on the glass tanks. Neutral density filters (248565, Grand Unified Optics, Beijing, China) were utilized to attenuate the intensity of the streetlights, thereby creating varying levels of nocturnal light intensities on the water surface, while ensuring no alteration to the light spectrum. Three groups were thus set up which, respectively, represent the simulated living environment in three situations (Figure 1a) [29,30,31,32,33]. The first group was the control group, simulating a moon-transparent night (ALAN = ~0.1 Lux) [29,33]. The second group was stimulated continuously with low artificial light to simulate the marine environment under the influence of city lights (ALAN = ~300 Lux) [29,30,31]. The third group was given continuous intense artificial light stimulation to simulate the marine environment of direct irradiation near industrial port facilities (ALAN = ~600 Lux) [32,33].

During the experiment, dead sea urchins (if any) were removed and recorded every day. Water quality assessments were performed at weekly intervals utilizing a portable water quality monitoring meter (YSI, Yellow Springs, OH, USA) (Appendix A). Six weeks later, the three groups of sea urchins that had been subjected to different long-term artificial light treatments were used for subsequent behavioral and physiological analysis.

### 2.3. Behavioral Analysis

Righting behavior, covering behavior, and food consumption of *H. crassispina* were measured in glass tanks (long × wide × deep = 40 × 30 × 30 cm; 20 cm of aeration water depth). Righting behavior [16,20,21,22,23]: A sea urchin was placed at the tank’s center with its aboral side facing downward, and the time taken for it to reorient itself to the oral side facing downward was measured as the righting response time. If the sea urchin failed to reorient within 10 min, the righting response time was recorded as 600 s. This experimental protocol aligns with previous studies on covering behavior [23,24,25,27]: A sea urchin was carefully positioned at the central point of the tank, with *U. Lactuca* available ad libitum at the bottom. The time required for the sea urchin to cover its body with *U. Lactuca* was recorded as the covering response time. If the individual failed to engage in covering behavior within a 30-min timeframe, its covering response time was recorded as 1800 s (Figure 1b). Food consumption [16,20,21]: Four sea urchins were delicately placed at the center of the tank, with weighted *U. Lactuca* available ad libitum at the bottom. The sea urchins began to feed about two hours later. Food consumption was quantified by subtracting the remaining uneaten quantity of *U. Lactuca* in the tank after a 24-h period from the initially supplied amount. To mitigate observer bias, a double-blinded approach was employed for the recording and analysis of all behavioral data.

### 2.4. Sampling Protocol

The sampling procedure of *H. crassispina* was conducted by the method described in previous papers [34]. Test diameter, test height, and Aristotle’s lantern length were assessed with a vernier caliper, with measurements recorded to the nearest 0.1 mm (CD-AX/C, MITUTOYO, Kawasaki city, Japan). Test weight, Aristotle’s lantern weight, gonads weight, and gut-weight were measured to the nearest 0.01 g by an electronic balance (C3-6553, AS ONE, Osaka, Japan). Subsequently, coelomic fluid was extracted and subjected to centrifugation at 10,000 rpm for 10 min in a refrigerated centrifuge (4 °C). The resulting supernatant was meticulously collected and preserved at −80 °C for subsequent analysis of monoaminergic activity. Tubular feet were cut off and washed thoroughly with ice-cold artificial coelomic fluid (10 mM CaCl_2_; 14 mM KCl; 50 mM MgCl_2_; 398 mM NaCl; 1.7 mM Na_2_HCO_3_; 25 mM Na_2_SO_4_) [35], and frozen quickly in liquid nitrogen for gene expression analysis. Test height/test diameter [25] and gonads weight/test weight (GSI) [20] were subsequently calculated.

### 2.5. Physiological Analysis

The concentrations of 5-hydroxytryptamine (5-HT) and 5-hydroxyindoleacetic acid (5-HIAA) in body cavity liquid samples were measured by commercial enzyme-linked immunosorbent assay (ELISA) Assay Kits (Nanjing Jiancheng Bioengineering Institute, Nanjing, China) according to the manufacturer’s instructions. These ELISA Assay Kits have been validated in marine life previously [36,37,38,39]. *Pax6* gene expression levels of tubular feet samples were measured using quantitative real-time polymerase chain reaction (qPCR) [40,41]. Total RNA was extracted using Trizol Reagent (Invitrogen, Carlsbad, CA, USA) containing bromochloropropane for phase separation. Afterward, RNA integrity was confirmed via electrophoresis on a 1.0% agarose gel, and quantification was conducted using a NanoDrop 2000 spectrophotometer (Thermo Scientific, Waltham, WA, USA). Subsequently, double-strand cDNA was synthesized from the total RNA template, and the construction of a cDNA library was achieved using the SMART cDNA construction kit (Clontech, Palo Alto, CA, USA). Primers for the study were designed utilizing Primer Premier 6 software (PREMIER Biosoft International, Palo Alto, CA, USA) and obtained commercially from Sangon Biotech (Shanghai, China). As a reference gene in sea urchins [42] (Table 1), the internal control gene *18S rRNA* was employed. Each qPCR reaction involved a 20 μL volume, including 10 μL of the 2× UltraSYBR Mixture, 0.2 μM specific forward and reverse primers, and 1.0 μL of the diluted cDNA template (10 ng/μL). The thermal cycling process was carried out on the ABI Prism 7900HT Sequence Detection System (PE Applied Bio-systems, Foster City, CA, USA), encompassing an initial denaturation step at 95 °C for 30 s, followed by 40 cycles of denaturation at 94 °C for 5 s, annealing at 60 °C for 30 s, and extension at 72 °C for 10 s. To confirm the specificity of amplification, a melting curve analysis of the PCR products was executed. Relative mRNA levels were determined using the 2^−ΔΔCt^ method [43]. The procedure of *Pax6* gene full-length conventional cloning of *H. crassispina* was fully described in the Appendix A: *Procedure* (*Sp*).

### 2.6. Data Statistics

Data throughout the text, tables, and figures are expressed as means ± standard error (S.E.). Statistical significance was established at *p* < 0.05. Data conforming to a normal distribution with equal variances underwent one-way analysis of variance (ANOVA) followed by Duncan’s multiple range post hoc test for statistical examination. When necessary, data were subjected to appropriate transformations, such as square root, exponential, or logarithm, to meet the assumptions of normal distribution and homogeneity of variance. Statistical analyses were conducted utilizing SPSS 17.0 for Windows (SPSS, IBM, Armonk, NY, USA, 2007).

## 3. Results

### 3.1. Survival

We found that ALAN at 0.1, 300, and 600 Lux could not cause sea urchin death. There was no significant difference in the number of surviving sea urchins among the individuals exposed to 0.1, 300, and 600 Lux of ALAN. However, sea urchins exposed to ALAN at 600 Lux showed severe spine loss during the experiment.

### 3.2. Righting and Covering Behavior

We found that ALAN at 600 Lux significantly increased the righting response time of *H. crassispina* compared to those exposed to ALAN at 0.1 and 300 Lux (*p* < 0.005; *p* < 0.005). However, there was no significant difference between the individuals exposed to 0.1 and 300 Lux of ALAN (*p* = 0.313; Figure 2a). Similarly, ALAN at 600 Lux significantly increased the covering response time of *H. crassispina* compared to those exposed to ALAN at 0.1 and 300 Lux (*p* < 0.005; *p* < 0.005). However, there was no significant difference between the individuals exposed to 0.1 and 300 Lux of ALAN (*p* = 0.442; Figure 2b).

### 3.3. Food Consumption

We found that the food consumption of *H. crassispina* significantly decreased after 6 weeks of ALAN exposure. Sea urchins exposed to ALAN at 300 Lux and 600 Lux resulted in a significant decrease in food consumption compared with the control (0.1 Lux) (*p* < 0.05; *p* < 0.05). However, there was no significant difference between the individuals exposed to 300 and 600 Lux of ALAN (*p* = 0.558; Figure 3).

### 3.4. Body Size

We found that ALAN at 600 Lux significantly decreased the body weight of *H. crassispina* compared to those exposed to ALAN at 0.1 Lux (*p* < 0.05; Figure 4a). However, there was no significant difference between the individuals exposed to 0.1 and 300 Lux or 300 and 600 Lux of ALAN (*p* = 0.133; *p* = 0.245; Figure 4a). In addition, there was no significant difference in test height, test diameter, and test height/test diameter among the individuals exposed to 0.1, 300, and 600 Lux of ALAN (Figure 4b–d).

### 3.5. Lantern Weight and Length

We found that Aristotle’s lantern weight of *H. crassispina* significantly decreased exposure to ALAN at 300 Lux and 600 Lux compared to those exposed to ALAN at 0.1 (*p* < 0.05; *p* < 0.05). However, there was no significant difference between the individuals exposed to 0.1 and 300 Lux (*p* = 0.906; Figure 5a). In addition, Aristotle’s lantern length among the individuals exposed to 0.1, 300, and 600 Lux of ALAN showed no significant difference after 6 weeks of the experiment (Figure 5b).

### 3.6. Gonad and Gut-Weight

We found that there was a significant difference in GSI (gonads weight/test weight) among the individuals exposed to 0.1, 300, and 600 Lux of ALAN (all *p* < 0.05; Figure 6a). There was no significant difference in gut-weight among the individuals exposed to 0.1, 300, and 600 Lux of ALAN (Figure 6b).

### 3.7. 5-HIAA/5-HT Ratio

We used the 5-HIAA/5-HT ratio as an indicator of serotonin system activity, and we found that ALAN at 600 Lux significantly increased the 5-HIAA/5-HT ratio of *H. crassispina* compared to those exposed to ALAN at 0.1 and 300 Lux of ALAN (*p* < 0.005; *p* < 0.005). However, there was no significant difference between the individuals exposed to 0.1 and 300 Lux of ALAN (*p* = 0.054; Figure 7).

### 3.8. Pax6 Gene Expression

We found that *Pax6* gene expression was significantly affected by ALAN at 600 Lux in *H. crassispina* compared with the control (0.1 Lux) (*p* < 0.005; *p* < 0.005). No significant difference in *Pax6* gene expression was found between the individuals exposed to 0.1 and 300 Lux of ALAN (*p* = 0.192; Figure 8).

## 4. Discussion

In this work, we constructed small-scale systems to investigate the effects of artificial light at night (ALAN) on fitness-related traits of *H. crassispina*. The main results were as follows: (1) ALAN at 600 Lux significantly increased the righting response time, the covering response time, and the 5-HIAA/5-HT ratio, and decreased food consumption, body weight, Lantern weight, GSI, and *Pax6* gene expression; (2) ALAN at 300 Lux significantly decreased food consumption, Lantern weight, and GSI.

### 4.1. Effects of ALAN on Behavioral Responses

Complex marine benthic environments shape a number of ecologically important behaviors in sea urchins. Covering and righting are two representative behaviors with fitness importance in antipredation [21,22,23,24,25]. The diminished covering and righting capacities of *H. crassispina* observed in this study suggest that extended exposure to ALAN might unfavorably impact fitness-related behaviors. This behavioral shift was linked to the decrease in sea urchin populations, potentially attributed to heightened predation pressure within the coastal ecosystem [28]. Similar outcomes highlighting the influence of ALAN on fitness-related behaviors have been documented in diverse coastal organisms. For instance, sea turtles (*Caretta caretta* and *Chelonia mydas*) may fail to navigate toward the ocean post-hatching due to ALAN-induced disruptions in their behavioral orientation [44]. Seabird migratory patterns also exhibit alteration, as birds are drawn to and linger around artificial light sources [45,46,47]. The composition of shallow water fish species underwent modification under ALAN influence: large predatory and small shoaling fish congregated around illuminated zones during the night, leading to reduced prey availability [30]. Notably, a study involving mud snails (*Hydrobia ulvae*) on French mudflats unveiled a decline in the crawling activity (and growth) of snails in response to light exposure [48].

The formation of metabolites like 5-HIAA arises subsequent to the re-uptake of 5-HT from the synaptic cleft, and their accumulation in neural tissue likely exhibits a time-dependent pattern [36]. Consequently, evaluating tissue concentrations of neurotransmitter metabolites fails to provide an immediate depiction of neural activity. As such, the ratio of metabolite to monoamine was adopted as a marker indicating heightened monoamine utilization. In this study, the 5-HIAA/5-HT ratio in *H. crassispina* exhibited an increase following prolonged ALAN exposure. This observation suggests that the serotonin system was stimulated in response to light pollution, causing a progressive augmentation in metabolite levels over time. Numerous reports have delved into serotonin’s influence on animal behavior, consistently revealing its inhibitory effects on traits such as aggressive behavior, feeding patterns, and motor activity [36,49]. Therefore, alterations in the concentration of the monoamine transmitter serotonin may underlie the internal mechanism driving the behavioral shifts observed in sea urchins within this study.

### 4.2. Effects of ALAN on Growth

*H. crassispina* subjected to prolonged artificial light exposure displayed a pronounced reduction in food consumption compared to those shielded from ALAN. This correlation is substantiated by our finding that ALAN significantly suppressed Aristotle’s lantern reflex, a pivotal factor in sea urchin food intake. This diminished food consumption provides a plausible explanation for the subsequent reduction in body size. These findings strongly indicate that light pollution could exert an impact on the growth of intertidal organisms. Correspondingly, studies by *Luarte* et al. involving both laboratory and field investigations demonstrated that the foraging behavior and growth rate of the amphipod *Orchestoidea tuberculate* were significantly affected by ALAN [50]. Field observations indicated that lower light levels (60 Lux) curtailed amphipod feeding and growth rates. The growth rates of amphipods under ambient conditions (darkness) were nearly three times higher compared to those exposed to ecological light pollution. Similarly, under ambient conditions, the food consumption rates were nearly two times higher than under exposure to ecological light pollution. The potential influence of light pollution on growth and development also extends to mollusks. Research unveiled a correlation between daily light exposure duration and the growth pace and stored energy levels in the pulmonate *Lymnaea stagnalis* [51]. These outcomes find alignment with investigations on rodents, specifically *Myotis emarginatus*, showcasing that light pollution diminishes consumption rates and impedes the growth of juvenile and suckling bats [52]. Thus, the decreased food consumption stemming from light pollution is likely to curtail the fitness of littoral organisms by reducing energy reserves.

### 4.3. Effects of ALAN on Reproductive Performance

Among the pivotal factors influencing sea urchin reproduction, food consumption holds paramount significance [20,21]. To gauge reproductive performance, we employed the Gonad-Somatic Index (GSI) and identified a GSI reduction in *H. crassispina* exposed to ALAN, which correlated with diminished food intake. This could suggest that sea urchins possibly allocate nutrients and energy to combat environmental stress, consequently compromising gonad tissue development. Furthermore, ALAN exhibited no significant influence on gut-weight in *H. crassispina*, a finding consistent with previous research [21]. This implies that sea urchin gonads are potentially more susceptible to environmental stressors than their gastrointestinal counterparts. Thus, ALAN exposure might detrimentally impact the reproductive capacity of sea urchins.

The development of gonad tissue follows a hormonal cascade, with gonadotropin-releasing hormone (GnRH) triggering the release of luteinizing hormone (LH) and follicle-stimulating hormone (FSH) from the pituitary gland, leading to the production of sex steroids in vertebrates [53]. The regulatory mechanism governing gonadal development is less defined in invertebrate sea urchins. As a result, we did not investigate relevant hormones to probe the mechanism through which light pollution affects sea urchin reproduction. Nonetheless, analogous studies have been conducted in marine fish. For instance, in female *Lateolabrax maculates*, mRNA expression of luteinizing hormone and follicle-stimulating hormone was suppressed even at white light levels as low as 1 Lux at night [54]. Similarly, a study in a natural setting found diminished mRNA expression of gonadotropins (luteinizing hormone and follicle-stimulating hormone) and circulating sex hormones (17β-oestradiol and 11-ketotestosterone) in European perch under street lighting conditions (13.3–16.5 Lux at the surface) [55]. Consequently, exploring the internal mechanism of light’s impact on gonad development in invertebrates such as sea urchins should be a prospective avenue of research.

### 4.4. Effects of ALAN on Pax6 Gene Expression

In a pioneering effort, we successfully isolated and sequenced a complete cDNA encoding a 440-amino-acid *Pax6* from the tubular feet of *H. crassispina*. We conducted an in-depth analysis of *Pax6*’s similarity and evolutionary relationships across various species (Appendix A), thereby furnishing fundamental biological insights for subsequent investigations into the *Pax6* gene within sea urchins. Notably, *Pax6* stands as a transcription factor gene with established involvement in eye development and photoreception in ocular-forming organisms [56]. The presence of *Pax6* expression signifies the preservation of photoreceptor cell specification processes, functioning as a canonical eye gene even in creatures devoid of specialized eyes [57]. Consequently, the *Pax6* gene emerges as a central focus for probing the photosensitive systems responding to the light milieu in sea urchins. For instance, the relative *Pax6* expression in *Strongylocentrotus intermedius* displayed a negative correlation with natural light intensity. However, *Pax6* expression patterns in *S. intermedius* underwent alterations following UV-B radiation exposure [58]. In line with this, our present experiment reveals that prolonged ALAN exposure also disrupts the typical expression of the *pax6* gene in *H. crassispina*. This suggests that persistent light stimulation might perturb molecules inherent to the photosensitive systems of sea urchins, ultimately yielding anomalous gene expression. On the whole, *Pax6* gene expression is poised to function as a sensitive gauge for assessing potential damage to the photosensitive system of sea urchins.

### 4.5. Sea Urchins in the Wild Light Environment

The profound impact of nocturnal light on the fitness traits of sea urchins can be attributed to their inherent nocturnal activity patterns [2]. Continuous exposure to light inhibits sea urchin activity, consequently slowing down behaviors such as covering and righting, as well as impairing their feeding efficiency. Additionally, this light exposure also leads to changes in hormone levels and gene expression within their bodies. This strong correlation between nocturnal light and the behavioral responses of sea urchins highlights the significant role that light plays in regulating their activity levels and overall ecological dynamics [31]. Despite this, considering the complexity of intertidal habitats, the impact of nocturnal light on sea urchins in the wild may not be consistent, especially in rocky coastal areas where cracks, rocks, and shade-providing seaweeds are present. Sea urchins may opt to inhabit shaded areas or regions with suitable light intensity. For instance, a study conducted on a drumfish species found that their activity range shifted in response to nighttime light exposure of 70 Lux [59]. However, the narrowing of the activity range was determined by the encroachment of intensifying nocturnal light into dark spaces. Therefore, it is crucial to adequately understand the complexity of natural settings. Further research conducted in the wild, including the use of large mesocosms, or in more natural laboratory settings, will be necessary.

## 5. Conclusions

These findings illuminate the potential detrimental impact of extended (6-week) exposure to ALAN on the behavioral responses, growth, reproductive performance, and photoreception of sea urchins. Importantly, the study findings reveal that the sea urchins’ responses to light are more pronounced when exposed to higher light intensities, emphasizing the increased threat posed by intensified artificial light at night (ALAN) to their overall fitness. This highlights the potential risks that intertidal coastal ecosystems face in areas characterized by prominent luminous installations, such as stadiums, ports, and wharves. These three locations, due to their excessive lighting, emerge as significant sources of disturbance and risk for the delicate balance of these ecosystems. Consequently, it is crucial to consider the conservation and management of these areas to mitigate the adverse effects of ALAN and preserve the ecological health and biodiversity of intertidal coastal environments. This study significantly advances our comprehension of the fitness-related attributes of sea urchins under ALAN exposure, furnishing crucial insights into the ecological risks of coastal regions.

## Figures and Tables

**Figure 1 animals-13-03035-f001:**
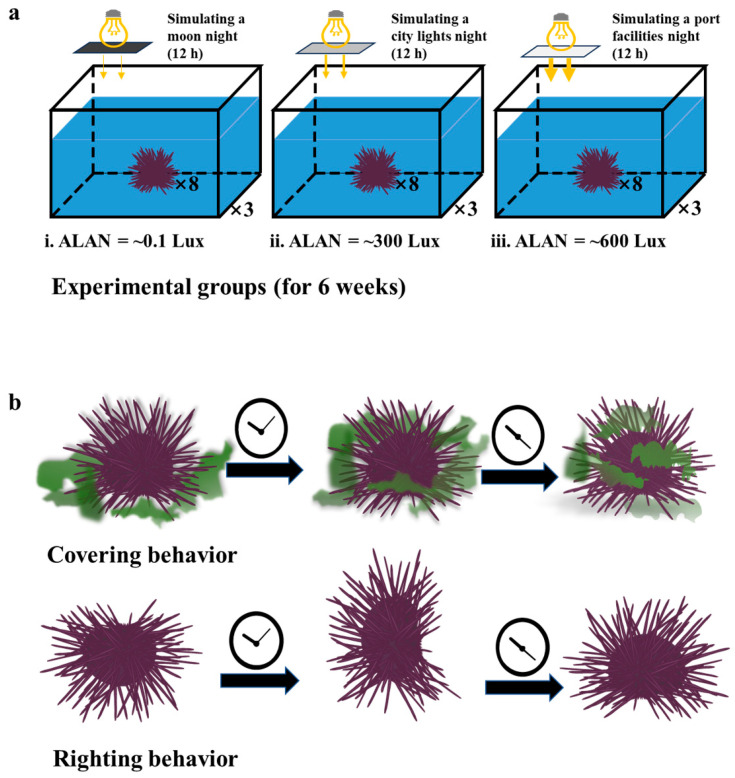
Schematic drawing of the three ALAN experimental groups (**a**) and the representative fitness behaviors (**b**) of sea urchins.

**Figure 2 animals-13-03035-f002:**
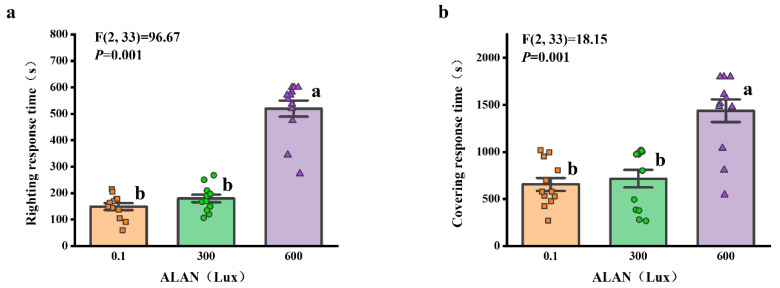
Effects of ALAN at different intensities for 6 weeks on righting response time (**a**) and covering response time (**b**) of sea urchins. Different letters indicate significant differences. Values are shown in data points and mean ± S.E. (*n* = 12).

**Figure 3 animals-13-03035-f003:**
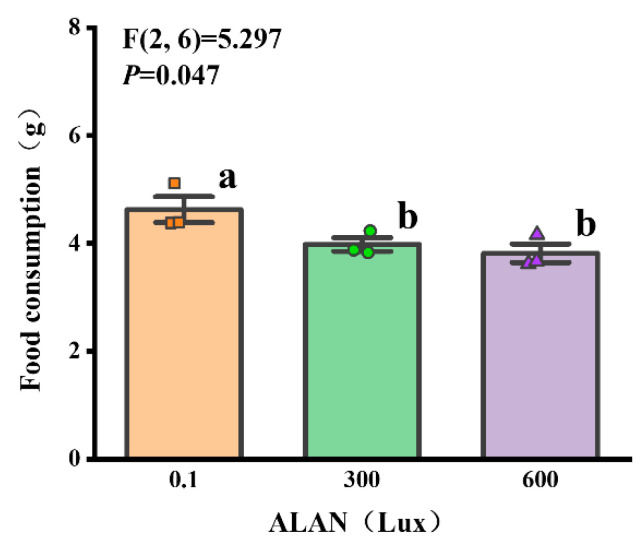
Effects of ALAN at different intensities for 6 weeks on food consumption of sea urchins. Different letters indicate significant differences. Values are shown in data points and mean ± S.E. (*n* = 3).

**Figure 4 animals-13-03035-f004:**
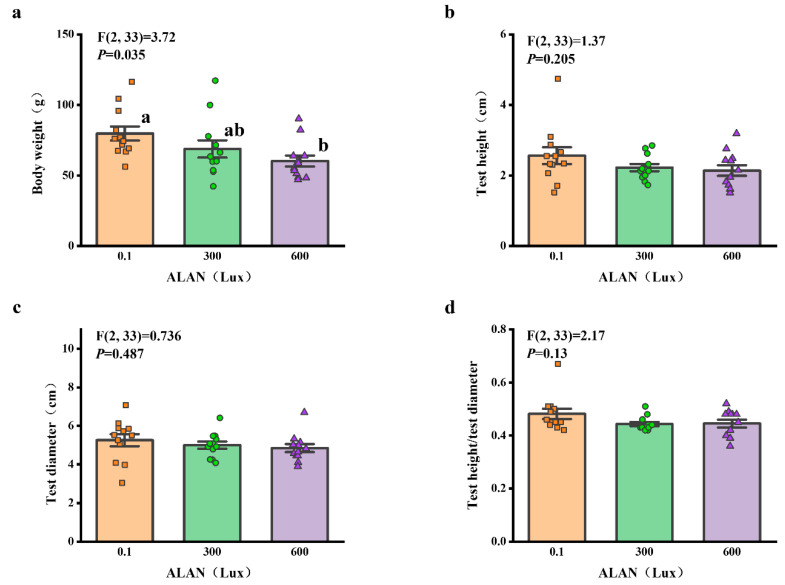
Effects of ALAN at different intensities for 6 weeks on body weight (**a**), test height (**b**), test diameter (**c**), and test height/test diameter (**d**) of sea urchins. Different letters indicate significant differences. Values are shown in data points and mean ± S.E. (*n* = 12).

**Figure 5 animals-13-03035-f005:**
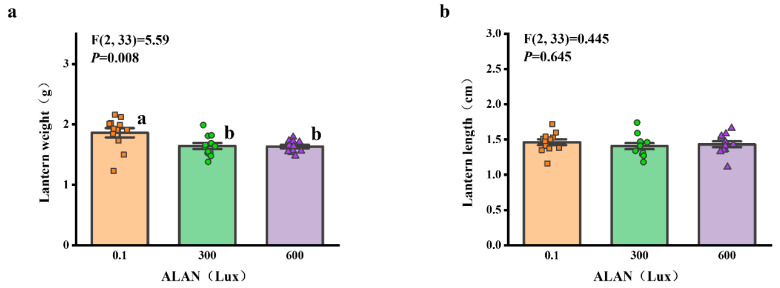
Effects of ALAN at different intensities for 6 weeks on Aristotle’s lantern weight (**a**) and Aristotle’s lantern length (**b**) of sea urchins. Different letters indicate significant differences. Values are shown in data points and mean ± S.E. (*n* = 12).

**Figure 6 animals-13-03035-f006:**
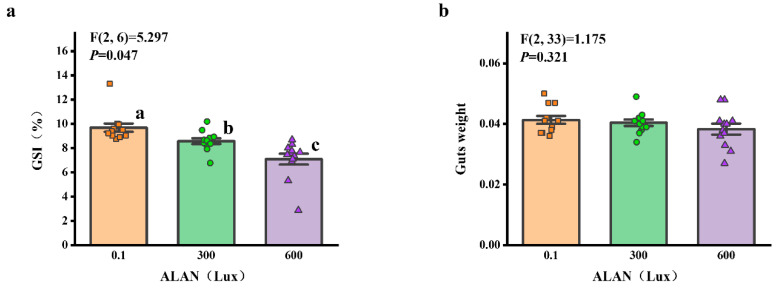
Effects of ALAN at different intensities for 6 weeks on GSI (**a**) and gut-weight (**b**) of sea urchins. Different letters indicate significant differences. Values are shown in data points and mean ± S.E. (*n* = 12).

**Figure 7 animals-13-03035-f007:**
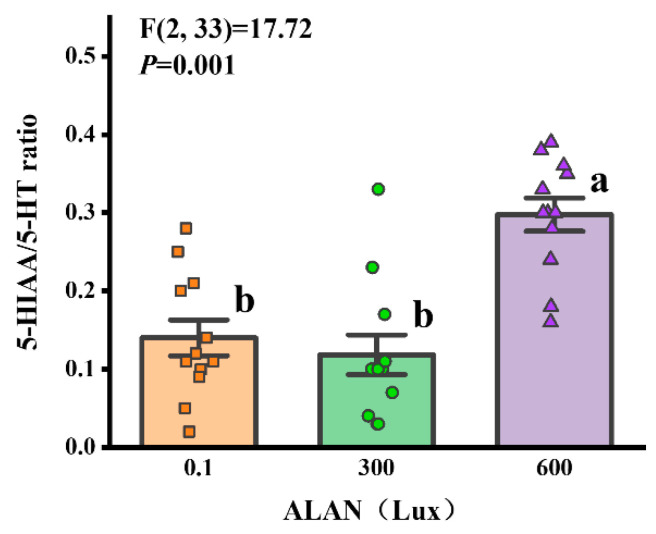
Effects of ALAN at different intensities for 6 weeks on coelomic fluid 5-HIAA/5-HT ratio of sea urchins. Different letters indicate significant differences. Values are shown in data points and mean ± S.E. (*n* = 12).

**Figure 8 animals-13-03035-f008:**
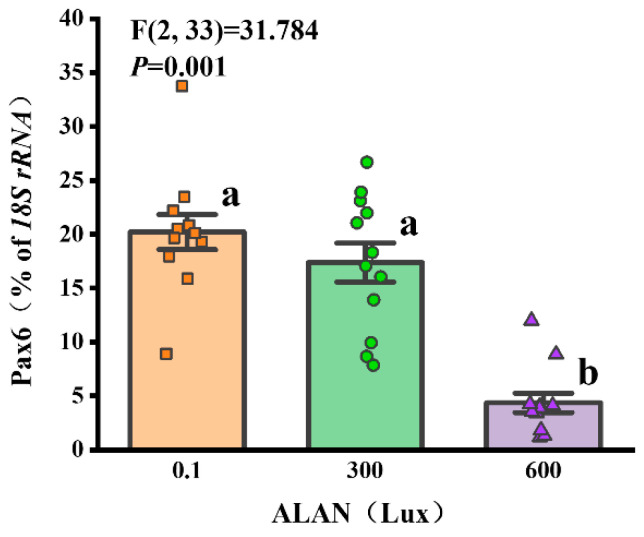
Effects of ALAN at different intensities for 6 weeks on *Pax6* gene expression of sea urchins. Different letters indicate significant differences. Values are shown in data points and mean ± S.E. (*n* = 12).

**Table 1 animals-13-03035-t001:** Nucleotide sequences of the primers for qPCR.

Primer Names	Sequence (5′→3′)	Application	AT (°C)
*Pax6*-F	AAGGCTGAAGATGATGAAGA	qPCR of *Pax6* gene	58
*Pax6*-R	GGAATGATTGGAAGACTGAC		
*18S*-F	ACGAAGGAGAAGACAAGG	qPCR of *18S rRNA* gene	56
*18S*-R	AAGCCACAAACGACAGTA		

AT: Annealing temperature.

## Data Availability

The data that support the findings of this study are available upon reasonable request from the corresponding author.

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
