# Peer review of "Effects of Artificial Light at Night on Fitness-Related Traits of Sea Urchin (Heliocidaris crassispina)"

_animals, 2023, doi:10.3390/ani13193035_

Round 1
Reviewer 1 Report
This study investigated the behavioral, morphological, and physiological responses of the sea urchin (Heliocidaris crassispina) after continuous exposure to ALAN. The findings revealed that continuous exposure to ALAN could cause an adverse effect on fitness-related traits including behavioral responses, growth, reproductive performance, and photoreception of sea urchins. This study contributes novel insights into the influence of light pollution on echinoderms. Overall, I have found the data presented in the current manuscript to be both intriguing and worthy of publication in Animals. Nevertheless, I would like to address several concerns and offer suggestions to assist the authors in further enhancing their manuscript.
Abstract:
1: “Our findings revealed that ALAN at 300 Lux substantially reduced food consumption, Lantern weight, and GSI.”
Comments: Although gonadosomatic index (GSI) is commonly used as an abbreviation for gonad development indicators, I would still recommend using the full name, 'gonadosomatic index,' for the first mention. This helps ensure clarity and understanding for readers who might not be familiar with the abbreviation.
Introduction:
2. Section 2: “Sea urchin Heliocidaris crassispina (previously Anthocidaris crassispina) is an echinoderm species found across a broad range from the intertidal zone to shallow waters in the Western Pacific region. Its habitat spans from the Japan Sea (at 38°N) to Hainan Island (at 19°N) [14-17].”
Comments: Please conduct a comprehensive analysis of the geographical distribution of the urchin population (Heliocidaris crassispina), incorporating precise and current references.
3. Section 2: “Health-related attributes, such as food consumption, body weight, and gut-weight, alongside reproductive traits like gonad weight, constitute foundational elements influencing the fitness of sea urchins.”
Comments: Make sure to use the term 'gut-weight' in the same way all throughout the text. This will make it easier for readers to understand and follow the results and discussions, making everything clearer and more connected.
Materials and Methods:
4: Section 4: “These ELISA Assay Kits have been validated in marine life previously [36].”
Comments: Please provide additional illustrative examples showcasing the applicability of ELISA Assay Kits in marine organisms. Additionally, can you provide detailed test methodologies that can serve as valuable references for researchers?
5. Section 4: “18S rRNA was applied as an internal control gene and has been validated as a reference gene in sea urchins [39].”
Comments: Make sure to put gene names in italics and please verify this in the entire text.
Discussion:
6. Section 2: “Correspondingly, studies by Luarte et al. involving both laboratory and field investigations demonstrated that the foraging behavior and growth rate of the amphipod Orchestoidea tuberculate were significantly affected by ALAN [47]. Field observations indicated that lower light levels (60 Lux) curtailed amphipod feeding and growth rates. The potential influence of light pollution on growth and development also extends to mollusks. Research revealed a correlation between daily light exposure duration and the growth pace and stored energy levels in the pulmonate Lymnaea stagnalis [48].”
Comments: Growth data can be quantified, and when reviewing similar studies, it is advisable to present as much specific detail as possible, rather than providing only general information. This part of the discussion needs to be improved.
Conclusions:
7. “Locations characterized by prominent luminous installations such as stadiums, ports, and wharves emerge as significant sources of risk for intertidal coastal ecosystems.”
Comments: This study investigated the effects of varying light intensities on the fitness-related attributes of sea urchins in three distinct coastal scenarios: moon-transparent night, city lights, and port facilities lighting. Please further strengthen and refine the discussion on these three scenarios.
References:
8. Please review the formatting of the references.
Reviewer 2 Report
The MS "Effects of artificial light at night on fitness-related traits of sea urchin (Heliocidaris crassispina) by Xu et al. is an original article that presents the study of light pollution in intertidal model organism H. crassispina. The MS is very well written and adequately presented to readers. The experimental design is well thought and implemented, although it does not seem to represent a real scenario where organisms are likely protected from ALAN from the complexity of the ecosystem (crevices, rocks, shade). Statistical analysis are described in sufficient detail and seem appropriate for the type of data presented. Several endpoints (behavioral, transcriptomics, fitness) were analysed and adequately presented/discussed. The discussion is objective and provides valid explanatory interpretations for the findings of the experiment. My opinion is that the MS is very good, and should be considered for publication in Animals after minor revisions/editing.
One question that could help improving the Discussion, and I found missing in the interpretation of the studies' findings, is related to how the nocturnal behaviour is influenced (or not) by light pollution. Considering sea urchin are generally nocturnal, how does light pollution during night time influence their behavioural response? I believe this simple explanation is missing, and could help explain some of the results presented (e.g impacts in feeding under high ALAN) and other data. It is likely that the urchins will be less active when exposed to continuous light. On the other hand, lower activity under ALAN could result in lower predation pressure in the environment.
Some reasoning about how the impact of similar ALAN levels in the environment should be added to the MS. Due to the complexity of intertidal habitats, specially in rocky shores, and considering the presence of crevices, rocks, seaweed that offer shade and shelter would mitigate some of the impacts of predicted in this study. The authors might consider adding a short paragraph to put the experiment into environmental context.
Minor comments:
- gene names (e.g (Pax6) could be presented in italic.
- Line84 is the 5000L tank is flow through? Please explain the flow rate.
-Line 87/122: urchin was fed ad libidum (not "liberally")
-Line97: please provide manufacturer's reference for Neutral density filters.
.line 135- electronic balance? or "electric"
line 144. avoid starting a sentence with a number
line 171: Why present SE instead of SD?
line288: increase instead of "elevation"
line 313: It is not clear why decreased body size negatively affects energy reserves. Energetic reserves can be proportional to body size. I suggest reviewing the sentence.
line354: Experiment instead of "experimentation"
Round 2
Reviewer 1 Report
The authors well addressed the problems I was concerned. I thus suggest the acceptance of the manuscript.